# Comparison of obstetric emergency clinical readiness: A cross-sectional analysis of hospitals in Amhara, Ethiopia

Kylie Dougherty[1]*, Abebe Gebremariam Gobezayehu[2,3‡], Mulusew Lijalem[2], Lamesgin Alamineh Endalamaw[2], Heran Biza[3‡], John N. Cranmer[3]

1 School of Nursing, Columbia University, NYC, NY, United States of America, 2 Emory-Ethiopia Country Office, Emory University, Addis Ababa, Ethiopia, 3 School of Nursing, Emory University, Atlanta, GA, United States of America

☯ These authors contributed equally to this work.
‡ AGG and HB also contributed equally to this work.
* kkd2131@cumc.columbia.edu

## Abstract

Measuring facility readiness to manage basic obstetric emergencies is a critical step toward reducing persistently elevated maternal mortality ratios (MMR). Currently, the Signal Functions (SF) is the gold standard for measuring facility readiness globally and endorsed by the World Health Organization. The presence of tracer items classifies facilities' readiness to manage basic emergencies. However, research suggests the SF may be an incomplete indicator. The Clinical Cascades (CC) have emerged as a clinically-oriented alternative to measuring readiness. The purpose of this study is to determine Amhara's clinical readiness and quantify the relationship between SF and CC estimates of readiness. Data were collected in May 2021via Open Data Kit (ODK) and KoBo Toolbox. We surveyed 20 hospitals across three levels of the health system. Commodities were used to create measures of SF-readiness (e.g., % tracers) and CC-readiness. We calculated differences in SF and CC estimates and calculated readiness loss across six emergencies and 3 stages of care in the cascades. The overall SF estimate for all six obstetric emergencies was 29.6% greater than the estimates using the CC. Consistent with global patterns, hospitals were more prepared to provide medical management (70.0% ready) compared to manual procedures (56.7% ready). The SF overestimate was greater for manual procedures 33.8% overall for retained placenta and incomplete abortion) and less for medical treatments (25.3%). Hospitals were least prepared to manage retained placentas (30.0% of facilities were ready at treatment and 0.0% were ready at monitor and modify) and most prepared to manage hypertensive emergencies (85.0% of facilities were ready at the treatment stage). When including protocols in the analysis, no facilities were ready to monitor and modify the initial therapy when clinically indicated for 3 common emergencies—sepsis, post-partum hemorrhage and retained placentas. We identified a significant discrepancy between SF and CC readiness classifications. Those facilities that fall within this discrepancy are unprepared to manage common obstetric emergencies, and employees in supply management may have difficulty

**Data Availability Statement:** The datasets can be found in Dans data repository at https://easy.dans.knaw.nl/ui/datasets/id/easy-dataset:305060

**Funding:** Author KD was funded under a training grant from the National Institute of Nursing Research (T32 NR007969) and is currently funded under an F31 predoctoral trainee grant (F31NR020569). These two grants support KD's tuition and education costs, and the National Institute of Nursing Research had no role in the study design, data collection, and analysis, decision to publish, or preparation of the manuscript. Data collection was supported by the Global Financing Facility; UNICEF-Ethiopia, but the Global Financing Facility did not play a role in study design, data analysis, decision to publish, or preparation of the manuscript.

**Competing interests:** The authors declare they have no competing interest.

**Abbreviations:** MMR, Maternal Mortality Ratio; MM, Maternal Mortality; WHO, World Health Organization; SSA, Sub-Saharan Africa; SF, Signal Function; CC, Clinical Cascades; BEmOC, Basic Emergency Obstetric Care; SRI, Service Readiness Index methodology; IV, Intravenous; MVA, Manual Vacuum Apparatus; EmOC, Emergency Obstetric Care; SSL, Saving Little Lives; ODK, Open Data Kit; OBGYN, Obstetrics and Gynecology.

identify the need. Future research should explore the possibility of modifying the SF or replacing it with a new readiness measurement.

## Introduction

Maternal mortality (MM) remains a significant public health concern [1]. The World Health Organization (WHO) reported in 2017, that 810 women die every day from preventable causes related to their pregnancy and or delivery [1]. One's country of residence is a profound determinant of MM. Ninety-four percent of all global maternal deaths occur in low-middle income countries (LMICs) [2]. Further, two-thirds of the global MM burden is borne by Sub-Saharan Africa (SSA) [1]. Ethiopia bears the fourth largest total number of maternal deaths in the world, with 14,000 annually [1, 2].

Furthermore, the region of Amhara consistently reports worse maternal outcomes and obstetric readiness compared to the country average [3, 4]. In a recent cross-sectional study, the authors found that 29.4% of mothers from Amhara had facility births, which is lower than the national rate of facility births of 38.9% [5]. Additionally, the 2018 Service Availability and Readiness Assessment found that Amhara was less prepared to administer parenteral antibiotics (44.0%), administer oxytocic drugs (47.0%), and performed assisted vaginal delivery (50.0%) when compared to Ethiopia as a whole (49.0%, 50.0%, 52.0% respectively) [6]. This is significant because those clinical skills are crucial for combatting the most common causes of MM.

### Obstetric emergency readiness

To have a profound impact on maternal outcomes Ministries of Health (MOH) can use specific and effective tools that are capable of accurately measuring a facility's clinical readiness to manage an obstetric emergency. This would provide the MOH with the ability to track the availability of adequate obstetric care. When the functional capacity of a facility is known, along with its weaknesses, interventions can be deployed to improve them. This will in turn increase the caliber of care being delivered and decrease MM. Obstetric emergency readiness at the facility level is defined as the "proportion of specified clinical items that are present at a facility on the day a facility inventory is conducted" [7]. This readiness can be evaluated as a whole, or researchers can look at readiness to manage individual obstetric emergencies. Both the Signal Functions (SF) tracer items and Clinical Cascades (CC) categorize readiness for the six most common obstetric emergencies into two different types of readiness. The two categories are *medical readiness* and *manual readiness*. *Medical readiness* included the ability to manage sepsis/infection, manage hemorrhage, and manage hypertensive emergencies. These conditions are defined as medical because they require some form of medication administration to treat the condition. The second category, *manual readiness*, includes the ability to manage retained placentas, incomplete abortions, and prolonged labor. These conditions are defined as manual because their treatment requires some form of manual action to treat the emergency.

### Signal functions (SF) for obstetric emergency readiness

The SF tool was created to provide succinct indicators of a facility's readiness to provide Basic Emergency Obstetric Care (BEmOC). It consists of three medical and three manual procedures that cover the care necessary to handle the six most common causes of MM [7, 8]. The most common global obstetric emergencies are hemorrhage (27.1%), infections or sepsis

(10.7%), pre-eclampsia and eclampsia (14.0%), incomplete abortion or ectopic pregnancy (7.9%), delivery complications and retained placenta (9.6%) [9]. In the WHO's Service Readiness Index methodology (SRI) specific items, also known as tracers, are used as proxies to objectively measure SF facility readiness for emergencies [10]. These tracer items are the core resources most essential for managing the emergency. The medical tracer items include three parenteral drugs (uterotonic, antibiotic, and anticonvulsant), three intravenous items (Intravenous (IV) solution, and a 2-part IV infusion kit (tubing and needle or cannula)), a manual vacuum apparatus (MVA), and two multi-purpose items (gloves and a light source) [7]. The SRI application of the SF uses these tracer items to create an overall emergency readiness indicator for facilities, countries, or regions based on the overall percentage of these items present at a facility on the day of assessment [6]. Table 1 maps the SF tracer items to the emergency it is used to treat.

The SF approach to using tracer items has emerged as the dominant approach for measuring BEmOC readiness at facilities around the world. This SF-based method is still the method recommended by WHO [7]. However, in recent years, researchers have called for improvements in the SF tracers or alternative methods for assessing BEmOC globally [7, 11–17]. First, the SF tracer item indicators have not yet been used to predict labor-related outcomes or a facility's practical readiness to identify and treat specific obstetric emergencies since the SRI readiness indicator is a pooled percent readiness for all six emergencies [11, 17]. Additionally, some studies have measured facility readiness using the SF tracers and other readiness tools and determined that a facility estimate of emergency readiness based only on SF tracer items consistently overestimates practical readiness [7, 10, 14]. If the SRI-SF estimates of facility readiness do overestimate a facility's practical readiness, it may make true readiness loss in a facility or region seem smaller than it actually is.

## Clinical cascade (CC) for obstetric emergency readiness

The CC for obstetric emergencies is an emerging set of readiness indicators designed in response to the apparent overestimates of emergency readiness from the commonly used SRI-SF tracer-based indicators [6]. The CC is a "clinically-oriented approach to measuring facility readiness" that measures the "step-wise cascading relationship between emergency resources [loss]" [10]. As with the SF tracer item readiness estimates, the CC uses an obstetric commodity inventory to estimate readiness. All SF tracer items are included in the cascade model. However, the CC adds a few additional durable goods, medications, and supplies that are critical to clinically manage each of the six common basic obstetric emergencies.

**Table 1. Tracer items and their corresponding obstetric emergencies.**

| Tracer Item | Obstetric Emergency[a] |
| --- | --- |
| IV Solution | 1, 2, 3, 4, 5, 6 |
| 2-Part IV Kit (tubing, needle/cannula) | 1, 2, 3, 4, 5, 6 |
| Parenteral Antibiotic | 1, 4, 5, 6 |
| Light Source | 4, 5, 6 |
| Parenteral Uterotonic | 2, 4 |
| Manual Vacuum Aspiration Kit | 5, 6 |
| Parenteral Anticonvulsant | 3 |
| Gloves | 2 |

[a]Obstetric Emergencies: (1) Sepsis-Infection, (2) Hemorrhage, (3) Hypertensive Emergency, (4) Retained Placenta, (5) Incomplete Abortion, (6) Prolonged Labor.

A unique feature of the CC is its attempt to measure the potential for providing quality care using commodities related to monitoring the initial treatment's efficacy and then adjusting-escalating treatment as indicated by the patient's initial response to therapy. To accomplish this, the CC reports the presence of resources, drugs, and emergency protocols required to adjust therapy if the patient does not successfully respond to initial treatment. For example, for a facility to be classified as ready to monitor and modify care for maternal hemorrhage, the facility would need a sphygmomanometer, stethoscope, uterotonic, urinary catheter, oxygen, and a hemorrhage protocol. While the presence of protocols and resources for escalating treatment does not guarantee the level of clinician skill, and the absence of protocols does not guarantee a lack of clinician ability, measuring these resources for monitoring-modifying therapy provides a commodity-based, readily measurable approach to estimating the quality of obstetric emergency care.

## Clinical logic for the clinical cascades

The CC categorizes emergency obstetric care (EmOC) readiness into three phases, identification of the emergency, treatment of the emergency, and monitoring and modification of treatment as clinically necessary [7]. Within this measurement, there is also a scale to determine readiness for each of the six most common obstetric emergencies individually. For a facility to be deemed ready to manage an obstetric emergency it must have all the necessary supplies to identify, treat, and monitor care for the specific obstetric emergency. The purpose of this study is to quantify facility readiness for BEmOC in Amhara, Ethiopia as measured by the SF tracer items and CC.

## Materials and methods

### Study design, setting and sample

This is a prospective, cross-sectional, facility-based analysis of basic obstetric emergency clinical readiness in Amhara, Ethiopia. Emergency-specific variables were identified from twenty hospitals as part of the national "Saving Little Lives" program being implemented in the region (SLL) [18]. SLL is an evidence-based intervention led by the Ethiopian government targeting child survival, specifically the key drivers of mortality for small babies, including respiratory distress, infection, and birth asphyxia [19]. SLL is a multi-year project with a targeted population of 290 hospitals in four regions in Ethiopia. Emory University is one of the collaborating partners in this project. To obtain our sample of hospitals, the team used the same included twenty hospitals that were already participating in SLL's year one work in Amhara.

### Study assumptions

While the use of protocols within the monitoring and modification phase of the CC assists in ensuring clinician knowledge to manage and monitor care following the treatment of an obstetric emergency, clinician skill was not measured. For this reason, a 100% skill level was assumed for the data analysis of CC readiness since skill level is not formally assessed in the SF readiness estimates or the CC model. Both approaches use commodities to estimate readiness.

### Ethics approval and consent to participate

This study was approved on December 13, 2021, by Amhara Public Health Institute and did not require individual consent because it was a system-level quality of care intervention. All methods were carried out in accordance with Amhara Public Health Institute guidelines and regulations. The study was exempt from Institutional Review Board oversight per Emory

University's review as a public health activity. The need for informed consent was waived by Emory's Institutional Review Board committee because this work was viewed as a system-level public health activity. There were no deviations from the study protocol after approval from Amhara Public Health Institute and Emory University's Institutional Review Board committee was obtained.

## Data collection

As part of baseline data collection for the SLL study, inventory data from twenty facilities was collected in May of 2021 in the Amhara region of Ethiopia. Facilities were co-selected for year one SLL interventions by the Emory-Ethiopia Partnership and Amhara Regional Health Bureau prior to this nested commodity-based study. The Emory-Ethiopia Partnership described the nature of the study, project objectives, and budget and the Regional Health Bureau assisted in identifying facilities from different hospital levels in the region that met project needs. Data were collected by trained data collectors on mobile tablets using Open Data Kit (ODK) [20]. The original database was designed using KoBo Toolbox [20]. The data were exported to STATA version 17 for analysis [21]. The twenty year one SLL hospitals represent the continuum of available BEmOC care in the region (primary, general, and referral hospitals) [22]. Select aggregate indicators were obtained from facility records (e.g., birth outcomes). Hospital leaders, including medical directors and labor and delivery unit heads, were interviewed to obtain facility-level information (e.g., target population, staffing). Permanent employees of the Emory-Ethiopia Amhara Regional Office in Bahir Dar were trained on data ODK and mobile tablet data collection procedures. All data collectors are health professionals, including clinical nurses and health officers, and gave technical input in the facility inventory data collection tools. Data collectors visualized the physical commodities used to create the SF- and CC indicators.

## Data analysis

Obstetric variables and facility characteristics were described with standard descriptive statistics (e.g., median and interquartile range for continuous variables that were not normally distributed or percentages for categorical variables). Readiness classifications were calculated using if-then logic. If then logic tabulates how many facilities have one specific medication or medical supply. If the facilities do have the item, the logic then takes that number and tabulates how many facilities met the criteria of having the original medication/medical supply plus an additional medication/medical supply. This logic is carried through the whole cascade from identification, to treatment, and monitoring and modification of an obstetric emergency. So, if a facility is missing an item in the identification phase, then they will not be included in the later calculation for readiness to treat or manage and modify care of an obstetric emergency. Five core readiness indicators were calculated.

   **1. Pooled mean readiness.**   Aggregate medical readiness (management of sepsis/infection, hemorrhage, hypertensive emergency) and manual readiness (management of retained placenta, incomplete abortion, prolonged labor) were reported as the pooled mean medical readiness and pooled mean manual readiness respectively for both SF tracer item method and CC method. Then the aggregate readiness across all six obstetric emergencies was reported as the overall pooled mean readiness using both tools.

   **2. Signal function and clinical cascade readiness discrepancy.**   Readiness classifications for each facility and each of the six most common obstetric emergencies were calculated using both the SF tracer item method and the CC. These readiness percentages determined the discrepancies between the two classifications by subtracting the pooled mean readiness for all

facilities per emergency using the CC method from the mean pooled readiness for all facilities per emergency using the SF tracer method (SF estimate [–] CC estimate = SF readiness overestimate).

**3. Readiness loss by sage of care.** Readiness loss along the stages of care, from identification, to treatment, and monitoring and modifying care, were calculated using the CC. Drop-offs in readiness between the stages of care were quantified as percentages (% of facilities ready to identify the emergency—% of facilities ready to treat the emergency = stage 2 readiness loss).

**4. Readiness loss across the treatment cascade.** Readiness loss across all three stages as a pooled mean loss was calculated using the CC. The mean drop-off in readiness for each individual emergency was quantified as a percentage ((stage 1 readiness loss + stage 2 readiness loss + stage 3 readiness loss)/3 = readiness loss across the treatment cascade).

**5. Impact of excluding protocols from stage 3 readiness calculations.** Facility readiness to manage the six most common obstetric emergencies using the CC method was calculated with the inclusion of the variable protocols and the exclusion of protocols in the readiness calculations. The difference in stage 3, management and monitoring care, was calculated between these two measurements to determine the significance of the presence of protocols on readiness scores (stage 3 readiness with protocols excluded–stage 3 readiness with protocols included = the impact of protocols on stage 3 readiness).

*Emergency readiness and clinical quality*. For this analysis, obstetric emergency readiness using the CC was calculated both with and without the inclusion of clinical protocols, during the monitor-modify phase since many facilities did not have protocols. For the primary analysis, we retained the protocols since this has been done in previous studies. Estimates of monitor-modify readiness without the protocols are reported as a secondary analysis (see Fig 4).

## Results

### Facility characteristics

The twenty hospitals in Amhara involved in year one study activities for SLL were included in the data collection for facility readiness. Data collection occurred between May 17 to the 27 in 2021. All facilities reported they were capable of providing BEmOC services by SF standards. All hospitals reported they had the resources to perform blood transfusions and had performed a cesarean section within 6 months from the date of data collection. All hospitals reported having at least one anesthetist on staff, however, only two hospitals reported having an anesthesiologist on staff. Additionally, only seven of the sample hospitals reported having obstetrics and gynecology (OBGYN) personnel on staff, and 18 reported having an emergency surgeon on staff. Table 2 provides a description of the facility characteristics.

### Overall emergency obstetric resource availability

The facility surveillance identified several critical supplies for managing obstetric emergencies that were often missing from the facilities. For medication, parenteral forms of diazepam (55.0% of facilities missing), gentamicin (65.0%), and penicillin (95.0%) were commonly missing. Many facilities also did not have protocols for offering quality obstetric emergency care including protocols for hemorrhage (65.0%), retained placenta (85.0%), incomplete abortion (85.0%), and infection-sepsis (100%). Finally, common commodities, such as aseptic gloves were often out of stock, with 65.0% of the surveyed facilities not having any available on the day of the inventory. For a detailed breakdown of the availability of critical obstetric emergency resources see S1 Table.

**1. Pooled mean readiness.** *Signal Function estimates of emergency readiness*. The overall pooled mean estimate of readiness, as defined by the presence of SF tracer items for the six

**Table 2. Facility characteristics.**

| Characteristics | n = 20 | % |
|---|---|---|
| **Zone** | | |
| South Gondar | 5 | 25.0 |
| Bahir Dar | 3 | 15.0 |
| West Gojam | 7 | 35.0 |
| Awi | 5 | 25.0 |
| **Facility Type** | | |
| Primary Hospital | 15 | 75.0 |
| General Hospital | 2 | 10.0 |
| Referral Hospital | 3 | 15.0 |
| **Catchment Population** | | |
| 100,000–499,999 | 13 | 65.0 |
| 500,000–999,999 | 2 | 10.0 |
| 1,000,000–4,999,999 | 3 | 15.0 |
| $\geq$ 5,000,000 | 2 | 10.0 |

common obstetric emergencies was 92.9%. Using the SF model, facilities were least prepared to manage prolonged labor (90.0%), and most prepared to manage hypertensive emergencies (98.8%). Facilities were generally less prepared to manage manual emergencies (92.9% ready) compared to medical emergencies (95.3% ready). See S1 Table for individual emergency readiness using SF.

*Clinical cascade estimates of emergency readiness*. The overall pooled mean estimate of readiness using the CC model was 63.3%. Facilities were least prepared to manage retained placentas (30.0%), and most prepared to manage hypertensive emergencies (85.0%). Unlike the SF estimates, there were larger discrepancies between medical readiness (70.0%) than manual readiness (56.7%). See S2 Table for individual emergency readiness.

**2. Signal function and clinical cascade readiness discrepancy.** *Quantifying the relationship between SF and CC readiness estimates*. The SF tracer items model overestimated obstetric emergency readiness by 29.6% compared to the CC model. The discrepancy was smaller for medical readiness, with an overestimate of 25.3%. However, the SF overestimate of readiness for manual procedures was larger at 29.6%. For individual emergencies, the smallest discrepancy was with managing hypertensive emergencies (13.8%), and the largest was with managing retained placentas (60.7%). Fig 1 shows the pooled mean facility readiness estimates using both the SF and CC models for the six emergencies.

**3. Readiness loss by stage of care.** There was a consistent decrease in facility readiness from identification (stage 1) to treatment (stage 2) and monitoring-modifying therapy (stage 3) in all six obstetric emergencies. There was a substantive drop-off in readiness to manage and modify care compared to providing the first-line treatment (Treat, Stage 2 [–] Monitor-Modify Stage 3). No facility in this sample was prepared to perform stage three activities (monitor and modify care) for sepsis/infection, hemorrhage, and retained placentas. Only five percent of facilities could manage incomplete abortions, and 10% could manage hypertensive emergencies. Facilities were most prepared to monitor and modify therapy for prolonged labor, however, this was only 25.0% of the facilities. Consequently, the emergency the Amhara facilities were most prepared to respond to still had 75% of the sample facilities unprepared to manage prolonged labor. Across the six emergencies, the largest drop-off in readiness occurred between stage 2 (treatment) and stage 3 (monitoring and modifying care). All the facilities were prepared to identify hemorrhage (stage 1), but the entirety of this readiness was lost by

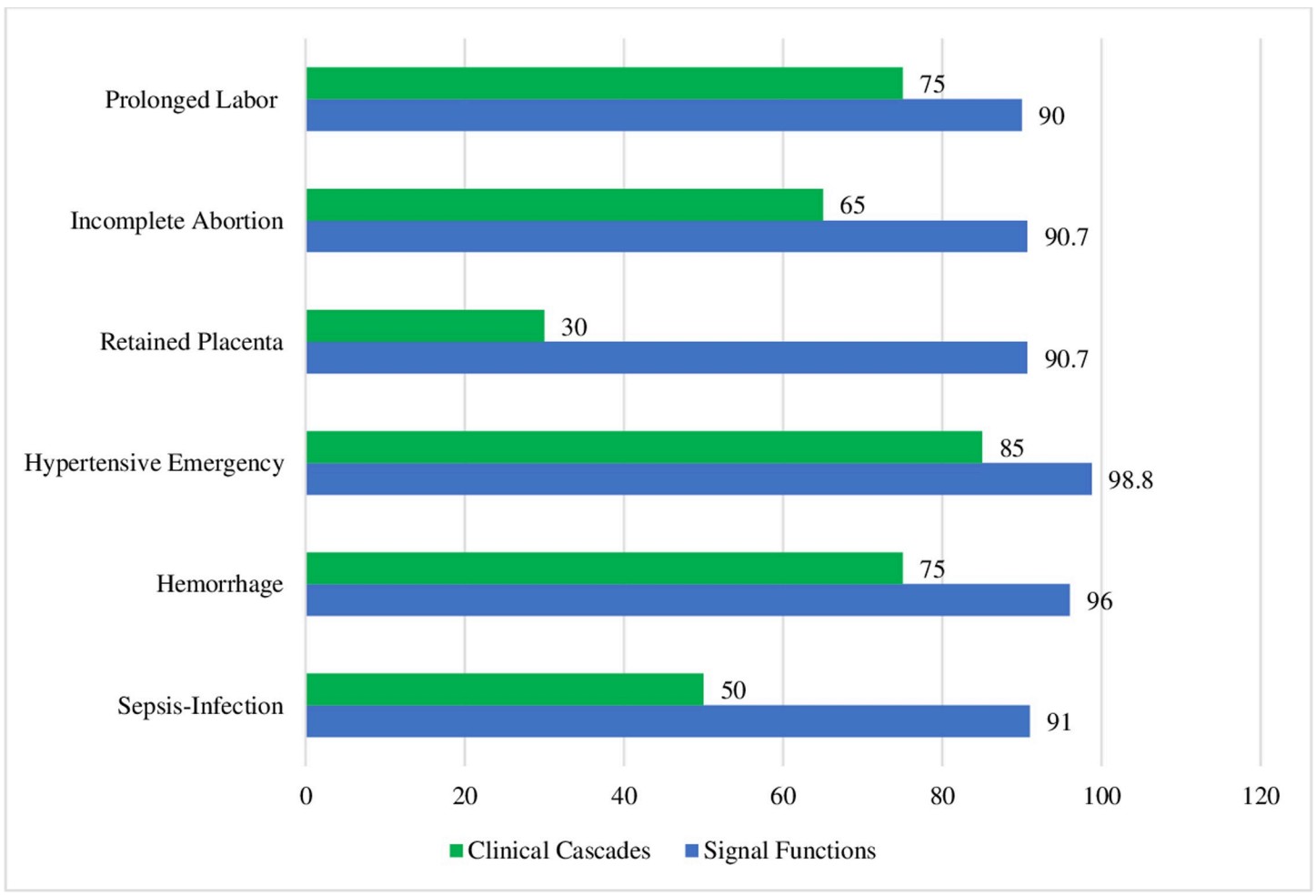

**Fig 1. Pooled percentage of facilities ready to manage obstetric emergencies using signal function and clinical cascade models.**

stage 3. Fig 2 shows the step-wise decrease in facility readiness through the treatment stages of obstetric emergencies, and Fig 3 depicts the increase in readiness loss across all six of the obstetric emergencies from identification of the condition to treatment, and monitoring and modification of care.

**4. Readiness loss across the treatment cascade.** This study revealed a pattern of a 30.7% drop-off in readiness across the six obstetric emergencies and stages of care (SD = 28.0). Hypertensive emergencies had the smallest drop in readiness across the cascade stages (M = 25.0%, SD = 22.9). The other 5 emergencies had mean drops ranging from 31.7%-33.3%. Table 3 displays the pooled mean drop-off across the three cascade stages of care and variability in the drop-off for each emergency using the standard deviation.

**5. Impact of excluding protocols from stage 3 readiness calculations.** While the inclusion of protocols in stage 3 CC calculations is critical to gauge the quality of care offered to mothers experiencing obstetric emergencies in our surveyed facilities, the authors were interested in exploring the impact these guidelines had on readiness estimates. Consequently, we conducted a sub-analysis of estimated readiness when the protocols were not used to measure a facility's readiness to monitor-modify therapy. In this sample of Ethiopian facilities, a large percentage of readiness was lost in stage 3 when readiness was based on the presence of

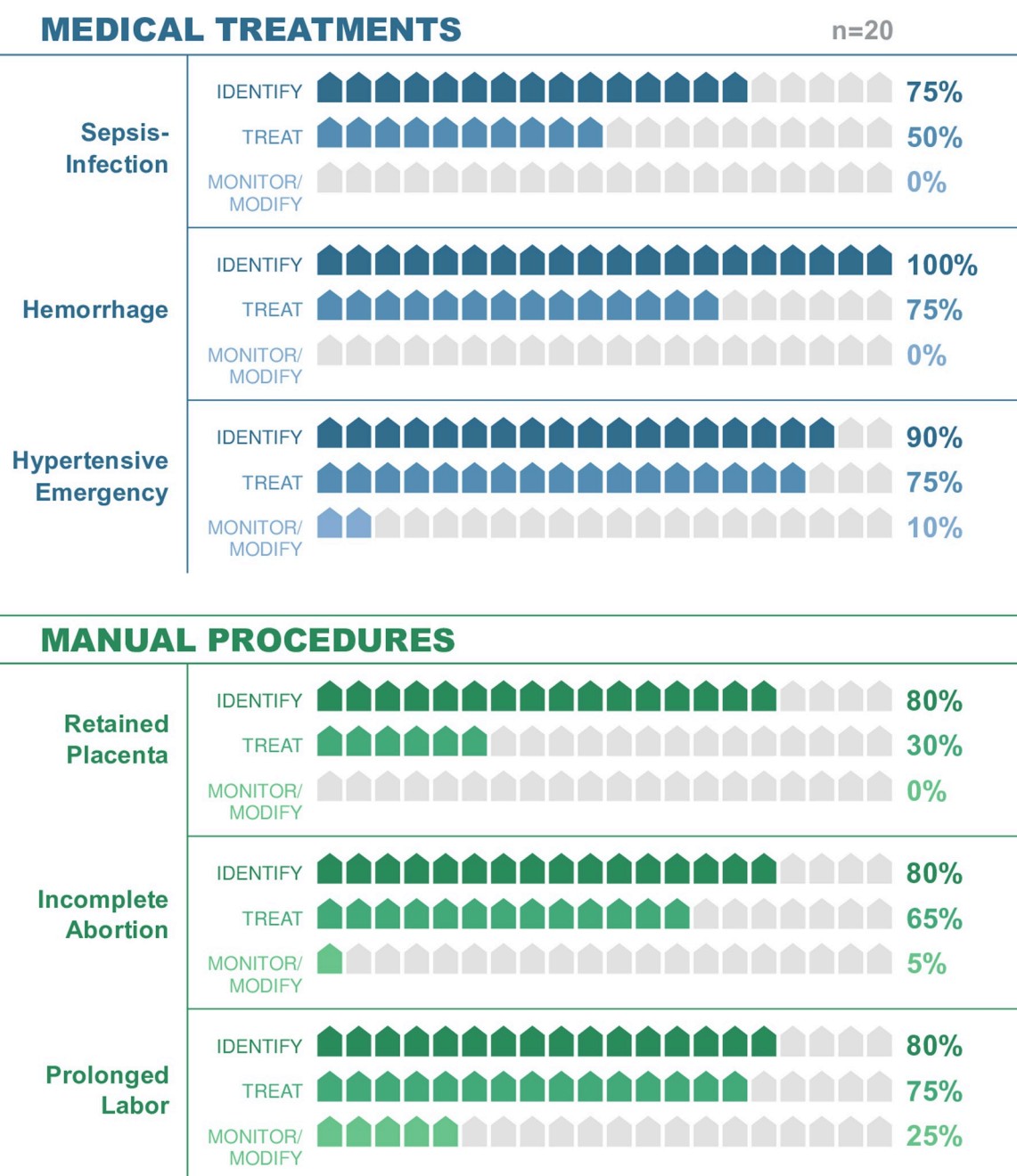

**Fig 2. Obstetric emergency readiness loss by stage of care: Clinical cascades readiness by stage.**

protocols. There is a 50.0% discrepancy in stage 3 readiness measured with and without protocols for the sepsis-infection cascade. With protocols, the readiness is 0.0% and without protocols, the readiness is 50.0%. However, for monitoring-modifying therapy for the retained placenta cascade, the discrepancy is much smaller at 5.0% (0.0% with protocol and 5.0% without). Fig 4 displays a comparison of facility readiness for each obstetric emergency when including and excluding protocols from CC calculations.

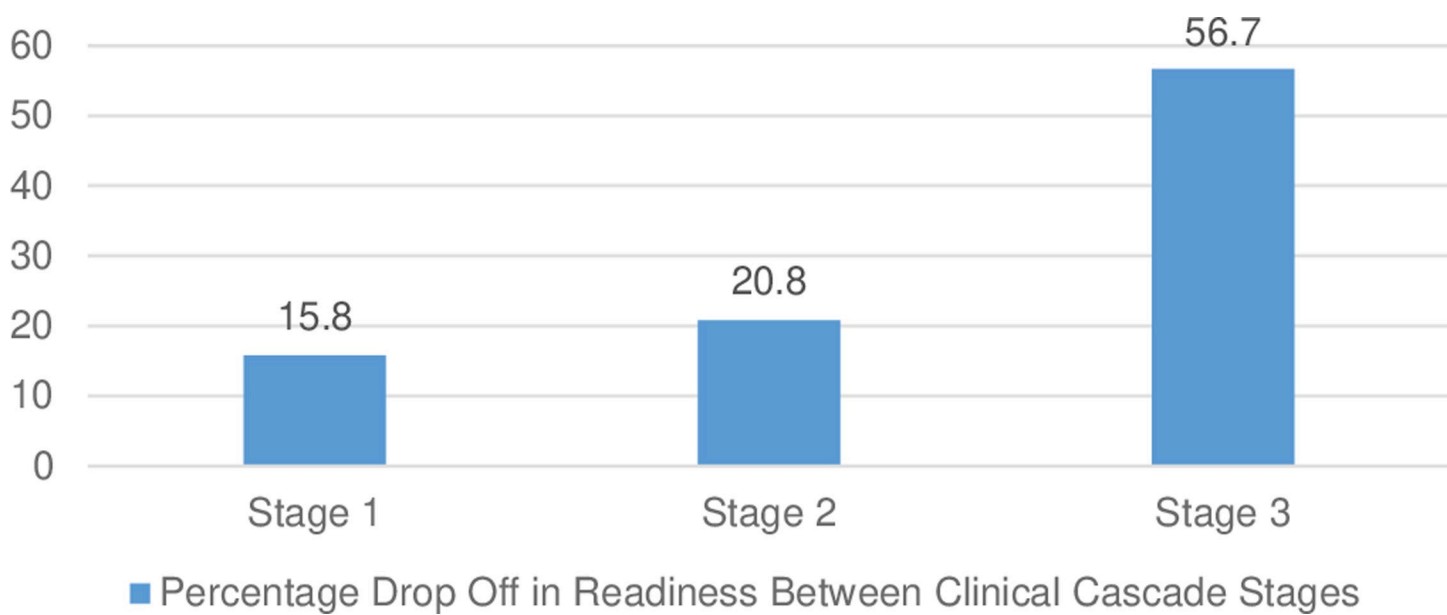

**Fig 3. Mean readiness loss between CC stages of care.**

## Discussion

### Implications for obstetric care

The discrepancy between the SF tracer items and CC can lead to the development of an "invisible need" where hospitals that are deemed ready to manage obstetric emergencies by the SF tracer items are in reality not ready to handle these cases. However, these facilities will not receive the necessary attention and resources they need because health planners and supply chain employees do not know that the problem exists. This is because federal and regional health planners lack critical inventory information that they need to make informed decisions on how best to use their resources and focus their attention. This can then lead to the primary global health metrics overestimating actual hospital readiness. Our findings align with previous research conducted in in Kenya and Uganda, which found SF tracer items overestimate facility readiness to manage obstetric emergencies.[7] Researchers and health planners should

**Table 3. Mean drop off in readiness by cascade across all 3 stages and among all facilities[a,b].**

| Clinical Cascade | Mean Drop Off Across 3 Cascade Stages | SD |
|---|---|---|
| **Sepsis-Infection** | 33.3% | 14.4 |
| **Hemorrhage** | 33.3% | 38.2 |
| **Hypertensive Emergency** | 30.0% | 39.1 |
| **Retained Placenta** | 33.3% | 15.3 |
| **Incomplete Abortion** | 31.7% | 24.7 |
| **Prolonged Labor** | 25.0% | 22.9 |
| **Mean of All Emergencies** | *30.7%[c]* | *28.0[d]* |

[a]n = 20 facilities

[b]The total number and percentage of facilities out of 20 that DO NOT have the capacity to fulfill these tasks

[c]Mean readiness drop off across 3 clinical cascade stages and 6 emergency cascades

[d]Standard deviations across 3 stages and 6 emergency cascades

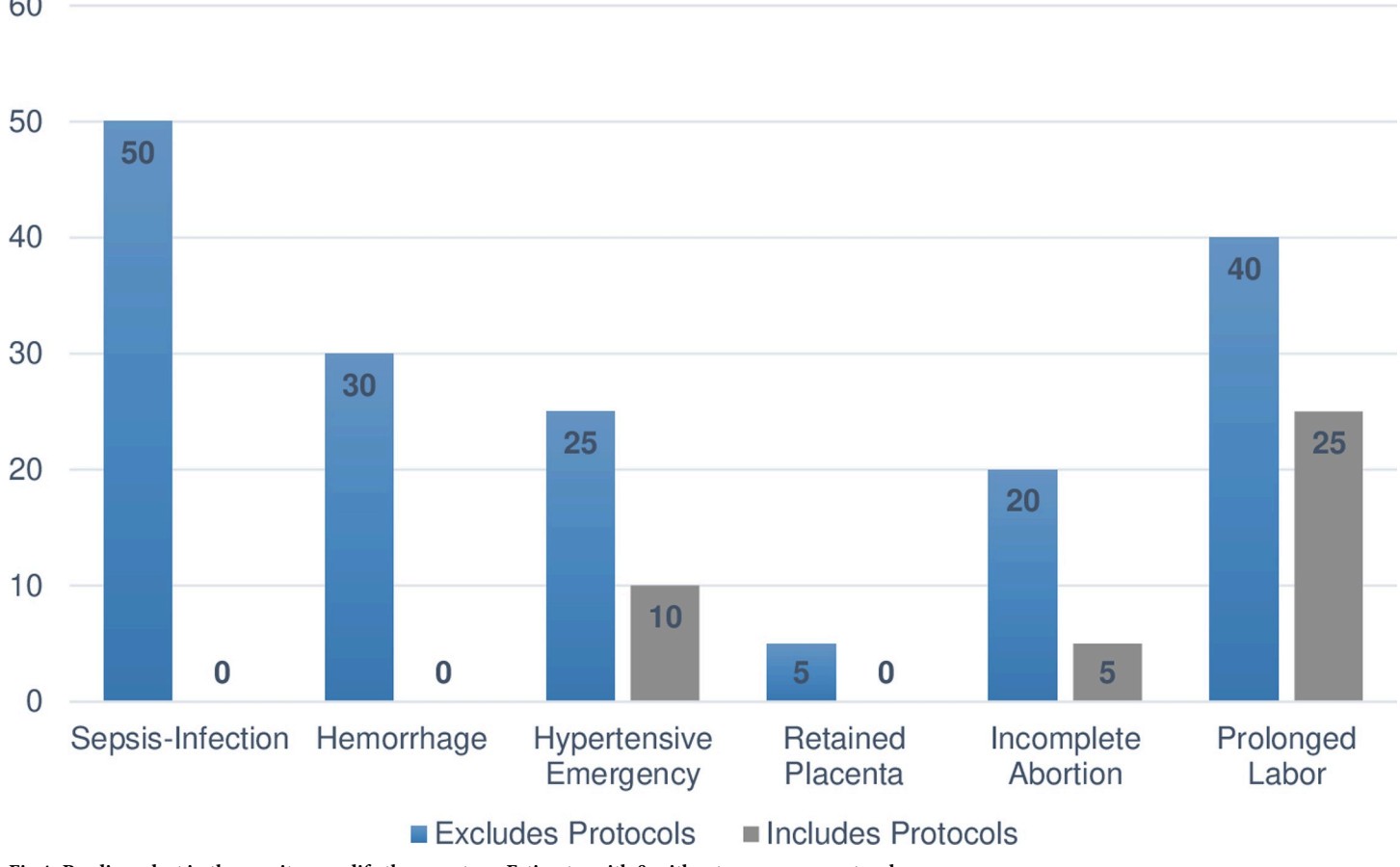

**Fig 4. Readiness lost in the monitor-modify therapy stage: Estimates with & without emergency protocols.**

consider modifying the SF tracer items or updating the indicator to prevent an overestimate of obstetric emergency readiness. Furthermore, future research should include outcome studies that explore the CC's ability to predict maternal morbidity and mortality rates.

## Implications for clinicians

On a local level, the continued use of the SF tracer items leads to the potential for facilities to be at risk of being unprepared to manage common obstetric emergencies. If the necessary supplies and medications are not available when a mother comes to the healthcare facility during an obstetric emergency then healthcare providers will not be able to provide high-quality care, even if they have the skills and clinical knowledge to do so. This lower-quality care can lead to worse health outcomes for the mother and the persistently high MM rates seen around the world, especially in areas such as SSA, including Ethiopia.

## Influence of protocols

As stated previously, the clinical cascades use the presence of protocols as a proxy for measuring the quality of care because it stands to reason that if there is a protocol available, then healthcare providers can use the protocol to guide their treatment even if they are unfamiliar with treating a particular obstetric emergency. As the analysis found, the presence of protocols can have a significant impact on readiness percentages, such as a 50% change in readiness for managing sepsis/infection. While the authors do not recommend removing protocols from

readiness classifications, it is important to explore their impact to gain a granular understanding of readiness. Additionally, this information can help to identify an area for readiness improvement that can significantly impact readiness classifications. For example, if the Amhara Regional Health Bureau were to focus on ensuring all their facilities that manage deliveries have up-to-date clinical protocols for managing obstetric emergencies, then providers would have the necessary resources to guide their care, and readiness scores based on the CC would increase for all six of the most common obstetric emergencies.

## Closing the gap in key commodity availability

Maintaining an accurate and real-time understanding of facility readiness to manage obstetric emergencies is critical to ensuring women receive quality care. This analysis was the first step in a larger study aimed at ensuring facility readiness to manage obstetric emergencies in Amhara, Ethiopia. This analysis helped to determine if accurate measurements are being used to determine readiness. The clinical cascades provide a detailed understanding of obstetric emergency readiness at any time point. The next step in this team's research in ensuring real-time understanding of readiness is through the use of health information technology that can track the presence of required medical supplies and medication. The use of electronic dashboards and integrated pharmaceutical logistics systems can track this information in real-time or closer to real-time, based on a location's technological capabilities, and close the gap in key commodity availability. The next step in these team' work is to develop electronic dashboards that will monitor facility readiness to manage obstetric emergencies in Amhara Ethiopia and alert regional health planners and inventory specialists when readiness is lost.

## Conclusion

The CC are an emerging method for measuring facility readiness to manage obstetric emergencies. The clinical cascades provide a clinically-oriented, granular approach to exploring facility readiness. By identifying the point along the treatment cascade regional health planners and policymakers can more accurately target interventions to ensure facilities are prepared for the most common obstetric emergencies. The average readiness classifications provided by the CC are also an efficient way to compare readiness in a region over time or compare readiness rates across geographical areas.

## Supporting information

**S1 Table. Availability of critical obstetric emergency resources.**
(DOCX)

**S2 Table. Comparison of emergency readiness using clinical cascades and signal functions, full sample.**
(DOCX)

## Author Contributions

**Conceptualization:** Kylie Dougherty, John N. Cranmer.

**Data curation:** Mulusew Lijalem, Lamesgin  Alamineh Endalamaw, Heran Biza.

**Formal analysis:** Kylie Dougherty, John N. Cranmer.

**Methodology:** Kylie Dougherty.

**Project administration:** Heran Biza.

**Writing – original draft:** Kylie Dougherty.

**Writing – review & editing:** Abebe Gebremariam Gobezayehu, Mulusew Lijalem, Lamesgin  Alamineh Endalamaw, Heran Biza, John N. Cranmer.

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
