## [Decision Letter · Decision Letter 0]

22 May 2023

PONE-D-23-05442Comparison of Obstetric Emergency Clinical Readiness: A Cross-Sectional Analysis of Hospitals in Amhara, EthiopiaPLOS ONE

Dear Dr. Dougherty,

Thank you for submitting your manuscript to PLOS ONE. After careful consideration, we feel that it has merit but does not fully meet PLOS ONE’s publication criteria as it currently stands. Therefore, we invite you to submit a revised version of the manuscript that addresses the points raised during the review process.

We look forward to receiving your revised manuscript.

Kind regards,

Abera Mersha, MSc.

Academic Editor

PLOS ONE

“Author KD was funded under a training grant from the National Institute of Nursing Research (T32 NR007969) and is currently funded under an F31 predoctoral trainee grant (F31NR020569). Data collection was supported by the Global Financing Facility; UNICEF-Ethiopia.”

5. Please ensure that you include a title page within your main document. You should list all authors and all affiliations as per our author instructions and clearly indicate the corresponding author.

Reviewers' comments:

Reviewer's Responses to Questions

**Comments to the Author**

1. Is the manuscript technically sound, and do the data support the conclusions?

Reviewer #1: Yes

Reviewer #2: Yes

2. Has the statistical analysis been performed appropriately and rigorously? 

Reviewer #1: Yes

Reviewer #2: Yes

3. Have the authors made all data underlying the findings in their manuscript fully available?

Reviewer #1: Yes

Reviewer #2: Yes

4. Is the manuscript presented in an intelligible fashion and written in standard English?

Reviewer #1: Yes

Reviewer #2: No

5. Review Comments to the Author

Reviewer #1: This is a good study, pointing out key factors of obstetric emergency clinical readiness. There are however some areas which I feel need attention as follows:

- editing of text- I realized your font varies in different sections

- Be clear on the number of emergencies on focus, is it 5 or 6? On methods you mentioned 5, and on results you mentioned 6

- On your key words, I think obstetric emergency should be added since it’s key in your title

- When starting a sentence, you put words I.e. line 5 of your introduction should not start with 94.0% but words

- Your list of abbreviations is incomplete- a number of abbreviations used in the document not included eg CEmOC, BEmOC, SF, SRI etc.

- Page I8, line 7- reasonable request. Be clear what you mean by tomorrow that.

Reviewer #2: Summary of research and overall impression

I am very impressed by this research because it is tackling the top causes of maternal mortality in Sub-Saharan Africa.

Strengths

1. The research is well detailed. The authors elaborated every detail of the study

2. The study presents an original work and meets the standards of research integrity

3. The study is paving way for an innovation in maternal and child health discipline which is uncommon. This is driving the future of research to problem solving rather than only academic arguments.

Minor issues

1. Some abbreviations are missing on the list of abbreviations yet not easily understood such as: BEmOC, SRI, SLL, SF

2. The manuscript needs to be proof read by English language expert for grammatical errors

3. Line 5-8 on page 3 are confusing. It talks about dividing most common obstetric emergencies into two forms of emergencies which are in turn listed as readiness in lines 6 and 7 of page 3. Clarify emergencies vs readiness.

4. There are abbreviations used which have not been defined anywhere in the manuscript: EmOC on line 8 of page 6, CEmOC on line 16 of page 10, IRB on line 11 of page 7, BEmONC on line 23 of page 7 and OBGYN on line 20 of page 10

5. The IRB number should be included if available under ethics approval section. This is to avoid doubts on approval status of the research

Major issues

1. The abstract does not have the findings of the second objective which was to quantify the relationship between SF and cascade estimates of readiness. A summary of major findings of each objective should appear under results in the abstract in the order of the objectives

2. Statement in line 33 on page one stated “but regional health planners are unable to identify the need” should be removed. This study did not determine the ability of regional health planners to identify facilities unprepared to handle obstetric emergencies. The conclusion should come from your key findings.

3. Give some brief introduction (in a line or two) about Saving little lives (SLL) program mentioned in lines 20 and 21 under materials and methods. It is mentioned as a study in line 15 of page 7 under data collection. Is this study part of an intervention or not? Clarify this.

4. What are twenty year one hospitals under data collection section? If they are the twenty hospitals where SLL program was being implemented in the first year, make it easy to understand. How long did data collection take? This should also clarify line 15 on page 10 under results section.

5. “These missing items highlight the critical need for accurate readiness monitoring to ensure that facilities do not run out of crucial supplies” on line 11 and 12 on page 11 of results should go to discussion section. Results are first presented without dissing them.

6. The results and data analysis sections should be presented in the order of study objectives. There is no heading for “quantifying the relationship between SF and cascade estimates of readiness” under results section. The discrepancies between SF and cascade estimates readiness can be stated under relationship between SF and cascade estimates of readiness

7. An upcoming innovation/innovation is mentioned in this study “electronic dashboards”. It is also important to state that this is a baseline study for the development of an innovation. This is to guide the reader further on the future of this research and the innovation to follow.

8. How do your findings relate with already existing similar researches in different parts of the world? This should be elaborated under discussion section

9. Author’s contribution section does not contain all the authors. Those who contributed to the work but do not meet the criteria for authorship can be mentioned in the Acknowledgments. (Refer to https://journals.plos.org/plosone/s/authorship#loc-authorship-requirements )

6. PLOS authors have the option to publish the peer review history of their article (what does this mean?). If published, this will include your full peer review and any attached files.

Reviewer #1: **Yes: **Dr Grace Danda

Reviewer #2: No

---

## [Author Response · Author response to Decision Letter 0]

30 Jun 2023

All reviewer comments are addressed in the document "Response to Reviewers."

---

## [Decision Letter · Decision Letter 1]

19 Jul 2023

Comparison of Obstetric Emergency Clinical Readiness: A Cross-Sectional Analysis of Hospitals in Amhara, Ethiopia

PONE-D-23-05442R1

Dear Dr. Dougherty,

We’re pleased to inform you that your manuscript has been judged scientifically suitable for publication and will be formally accepted for publication once it meets all outstanding technical requirements.

Kind regards,

Abera Mersha, MSc.

Academic Editor

PLOS ONE

Additional Editor Comments (optional):

Reviewers' comments:

Reviewer's Responses to Questions

**Comments to the Author**

1. If the authors have adequately addressed your comments raised in a previous round of review and you feel that this manuscript is now acceptable for publication, you may indicate that here to bypass the “Comments to the Author” section, enter your conflict of interest statement in the “Confidential to Editor” section, and submit your "Accept" recommendation.

Reviewer #1: All comments have been addressed

2. Is the manuscript technically sound, and do the data support the conclusions?

Reviewer #1: Yes

3. Has the statistical analysis been performed appropriately and rigorously? 

Reviewer #1: Yes

4. Have the authors made all data underlying the findings in their manuscript fully available?

Reviewer #1: Yes

5. Is the manuscript presented in an intelligible fashion and written in standard English?

Reviewer #1: Yes

6. Review Comments to the Author

Reviewer #1: All previous comments have been attended to except a few references which I still feel are outdated I.e. all those more than 10 years old.

7. PLOS authors have the option to publish the peer review history of their article (what does this mean?). If published, this will include your full peer review and any attached files.

Reviewer #1: **Yes: **Dr Grace Danda

---

## [Editor Report · Acceptance letter]

26 Jul 2023

PONE-D-23-05442R1 

Comparison of Obstetric Emergency Clinical Readiness: A Cross-Sectional Analysis of Hospitals in Amhara, Ethiopia 

Dear Dr. Dougherty:

I'm pleased to inform you that your manuscript has been deemed suitable for publication in PLOS ONE. Congratulations! Your manuscript is now with our production department. 

Kind regards, 

on behalf of

Mr. Abera Mersha 

Academic Editor

PLOS ONE